# The Potential for Fetal Alcohol Spectrum Disorder Prevention of a Harmonized Approach to Data Collection about Alcohol Use in Pregnancy Cohort Studies

**DOI:** 10.3390/ijerph16112019

**Published:** 2019-06-06

**Authors:** Nancy Poole, Rose A. Schmidt, Alan Bocking, Julie Bergeron, Isabel Fortier

**Affiliations:** 1Centre of Excellence for Women’s Health, Vancouver, BC V6H 3N1, Canada; Rose.Schmidt@cw.bc.ca; 2Lunenfeld-Tanenbaum Research Institute, Sinai Health System, Toronto, ON M5G 1X5, Canada; Alan.Bocking@sinaihealthsystem.ca; 3Department of Obstetrics and Gynaecology, Mount Sinai Hospital, Toronto, ON M5G 1X5, Canada; 4Department of Obstetrics and Gynaecology, University of Toronto, Toronto, ON M5G 1X5, Canada; 5Research Institute of the McGill University Health Center, Montreal, QC H3H 2R9, Canada; jbergeron@maelstrom-research.org (J.B.); isabel.fortier2@mcgill.ca (I.F.)

**Keywords:** prenatal alcohol exposure, cohort harmonization, FASD prevention

## Abstract

Prenatal alcohol exposure is a leading cause of disability, and a major public health concern in Canada. There are well-documented barriers for women and for service providers related to asking about alcohol use in pregnancy. Confidential research is important for learning about alcohol use before, during and after pregnancy, in order to inform fetal alcohol spectrum disorder (FASD) prevention strategies. The Research Advancement through Cohort Cataloguing and Harmonization (ReACH) initiative provides a unique opportunity to leverage the integration of the Canadian pregnancy and birth cohort information regarding women’s drinking during pregnancy. In this paper, we identify: The data that can be collected using formal validated alcohol screening tools; the data currently collected through Canadian provincial/territorial perinatal surveillance efforts; and the data currently collected in the research context from 12 pregnancy cohorts in the ReACH Catalogue. We use these findings to make recommendations for data collection about women’s alcohol use by future pregnancy cohorts, related to the frequency and quantity of alcohol consumed, the number of drinks consumed on an occasion, any alcohol consumption before pregnancy, changes in use since pregnancy recognition, and the quit date. Leveraging the development of a Canadian standard to measure alcohol consumption is essential to facilitate harmonization and co-analysis of data across cohorts, to obtain more accurate data on women’s alcohol use and also to inform FASD prevention strategies.

## 1. Introduction

Alcohol is a widely used substance among women of reproductive age in Canada [1,2,3,4]. Researchers have identified sex-specific effects of alcohol on women’s health and national guidelines related to low risk alcohol drinking reflect this evidence, including recommendations for no alcohol use in pregnancy [5,6].

Prenatal alcohol exposure (PAE) is a leading cause of disability and is considered to be a major public health concern in Canada [7,8]. Fetal Alcohol Spectrum Disorder (FASD) refers to the constellation of effects associated with PAE. These include a continuum of neurodevelopmental impairments, resulting in problems with learning, memory, attention, executive functioning, affect regulation, adaptive behavior, motor skills and social skills [9,10]. A wide range of long-term effects on health have been reported by adults with FASD, including, but not limited to: An increased risk of autoimmune disorders, hypertension, sleep disorders, osteoarthritis, vision and hearing problems, digestive problems, migraines and mental health concerns [11]. The prevalence of FASD in the Canadian population is estimated to be as high as 4% [12,13,14,15,16]. Alcohol use in pregnancy may also lead to preterm birth, defined as a delivery at less than 37 completed weeks of gestation, and the baby being small for gestational age [17]. Given the considerable negative health effects of heavy alcohol use on women’s health overall, and the risk of FASD when pregnant women consume alcohol, it is critical to understand the level and patterns of women’s drinking before, during and following pregnancy.

There are well-documented barriers for women, and for service providers, related to asking about alcohol use in pregnancy in the clinical context. Service providers report being reluctant to ask women about their alcohol use, due to fears of jeopardizing their clinical relationship, or being perceived as judging and shaming women’s behavior [18,19]. While research [20] and prevention efforts such as the Women Want to Know campaign in Australia (fare.org.au) indicate that women appreciate the opportunity to discuss their alcohol use, and learn about ways to improve their health, persistent barriers exist for women to answering questions about alcohol use during pregnancy. For example, women report not feeling safe to report their alcohol use for fear they will be seen negatively by their service provider for their continued use in pregnancy, out of concern about involvement from the child welfare or justice systems, or out of the perception that accessible and helpful outpatient options are unavailable [19,21,22].

As such, the confidential research context, in addition to the clinical context, could serve as an important setting for learning about alcohol use before, during and after pregnancy to inform FASD prevention strategies. In the confidential research context, the challenges mentioned above related to the safety of reporting of alcohol use are lowered. But, individual cohort studies generally include a limited number of regionally-distributed participants, and it is often essential to combine data across studies to truly unleash surveillance and research potential. An increasing number of national and international collaborative projects are opting for a co-analysis of individual participant data across studies to obtain sufficient statistical power, perform refined subgroup analysis, increase exposure heterogeneity, and enhance capacity to undertake comparison, cross validation or replication analysis [23,24,25]. The Lifecycle [26] and Enrieco [27] projects are examples of such international initiatives. Generally, these initiatives use existing data to support their research activities, but the heterogeneity of the data collected (different questions, categories, etc.) limits the capacity to harmonize data across studies. The use of compatible tools to prospectively collect common measures would facilitate such harmonization [28,29].

The Research Advancement through Cohort Cataloguing and Harmonization (ReACH) initiative (https://www.maelstrom-research.org/mica/network/reach#/) provides a unique opportunity to leverage the integration of Canadian pregnancy and birth cohort information regarding women’s drinking during pregnancy. At least 12 of the cohorts documented in the ReACH catalogue include information regarding women’s alcohol use during pregnancy. These cohorts are based in Alberta, Ontario, and Quebec, and are studying maternal and pregnancy health and risk factors such as nutrition, weight, smoking exposure, hypertension, mental health, stress, and various environmental exposures. These 12 cohorts combined include data on a total *n =* 23,856 mothers, and *n =* 17,877 children.

This article will explore how women are being asked information on alcohol use in pregnancy by cohort studies, and examine how a consistent approach that is informed by best practice in screening for alcohol use in clinical contexts can improve this data collection. Harmonization of data could overcome some existing challenges in Canadian FASD prevention research, which is currently limited by factors such as small data sets, and the lack of current population-based surveillance surveys that include information on maternal substance use.

A significant benefit of the cohort data available in the ReACH catalogue is that it includes birth outcomes, and longitudinal data collected about infants and children, some up to 12 years post-partum. Previous research has indicated that women who consume alcohol during pregnancy may not represent a homogeneous group [30,31,32]. The larger sample size and increased statistical power afforded by harmonization will allow for a more nuanced analysis, such as identifying differing trajectories of alcohol use before, during and after pregnancy, as well as the effects of these trajectories, and the relationship of social demographic characteristics on infant and child outcomes. Such data on women’s alcohol use will provide information which is difficult to obtain through clinical interventions, and will be highly useful to inform FASD prevention strategies. Identifying social demographic correlates and trajectories of alcohol use among Canadian women will allow for the development of more targeted FASD prevention efforts. However, leveraging the development of a Canadian standard to identify alcohol consumption by women is essential to facilitate the harmonization and co-analysis of data across cohorts.

## 2. Materials and Methods

In this paper we identify: (1) The data that can be collected from formal validated tools; (2) the approach to data collection from Canadian provincial/territorial perinatal surveillance efforts; and (3) the data collected in a research context from the pregnancy cohorts included in the ReACH Catalogue as of November 2018. We use these three sources to make recommendations on guidelines for data collection about women’s alcohol use by future pregnancy cohorts. As there is no gold standard for the best practice related to screening and brief intervention for alcohol use and alcohol use problems with pregnant women, we describe below the lessons from each of these sources, and make recommendations for a consistent approach to asking about alcohol in the research context.

### 2.1. Identification of Current Practices in Screening for Alcohol Use in Clinical and Surveillance Contexts

We present data on current approaches to asking about alcohol use in pregnancy from structured screening tools used to identify risky drinking developed by organizations such as the World Health Organization (WHO). The screening tools and approaches discussed were identified during a literature review conducted by the Centre of Excellence for Women’s Health in 2017 with the goal of summarizing existing evidence on brief interventions with girls and women, including strategies for engaging their partners and support networks [33]. The literature review included academic literature published between 2004 and 2017, identified using EBSCOHost Research Databases and grey literature (e.g., unpublished reports, practice guidelines) identified though targeted web searches [33].

We also present the screening questions used and documented on antenatal records by provinces and territories in Canada. In April 2018, an online meeting of Canadian perinatal data collection (surveillance) experts was sponsored by the Centre of Excellence for Women’s Health for the purpose of discussing data collection methods in use across Canada, and how these practices align with what is known about effective brief interventions with women who use substances [33]. We discuss these in the context of the evidence underlying Canada’s Low Risk Alcohol Drinking Guidelines [6].

### 2.2. Identification of Data Collected in the Research Context

In November 2018, there were twelve pregnancy cohorts in the ReACH Catalogue that included at least one variable on maternal alcohol use during pregnancy. The data labels and response categories for all alcohol use variables in these cohorts were extracted, and variables that were collected during seven data collection periods (during each of the three trimesters, birth, 1–6 months, 7–18 months and 1.5 to 3 years postpartum) were charted in an Excel™ (version 16, Microsoft, Redmond, WA, USA) spreadsheet. Three studies also include data on alcohol use at 3–6 years and one at 6–12 years postpartum, but this information was not included in the present analysis. Questions with different leading statements that collected the same data, using the same response categories, were considered the same question.

For example, questions such as “Since you were pregnant, how often did you drink?” asked during the first trimester were combined with “In the last three months, how often did you drink?” However, if similar questions had different response categories (e.g., the same question with different categories during pregnancy compared with postpartum), or included different periods of recall, they were counted as separate questions (e.g., “In the 12 months prior to your pregnancy, how often did you drink?” was considered separate from “Currently how often do you drink?”). The data labels and response categories were compared across the cohorts to describe the information collected by each study, and to identify specific questions used to gather information regarding the frequency, quantity and heavy alcohol use before, during and after pregnancy. Information about the actual questions used during data collection were available for seven of the 12 cohorts extracted from the ReACH Catalogue, while the remaining five cohorts provided only data labels with information about the variable content (e.g., “Frequency of red wine (1 glass = 4 oz.)”).

## 3. Results

### 3.1. Current Approaches to Screening for Alcohol Use in Clinical and Surveillance Contexts

Screening for and recording information about alcohol use with pregnant women and women of child-bearing age is seen to be a critical and practical process to identify women at risk of having a child with FASD. Structured screening tools have been recommended to identify alcohol use problems in the general population, and some have been designed for use with pregnant women [6,34].

#### 3.1.1. Structured Screening Tools to Identify Risky Drinking in General and with Pregnant Women

In research with pregnant women it is important to know about any use of alcohol (not only problematic use), and changes made in use over the pregnancy. The Alcohol Use Disorders Identification Test (AUDIT) [35] is a commonly used screening tool to assess alcohol consumption, drinking behaviors and alcohol-related problems in the general population. The full AUDIT questionnaire consists of 10 questions. The AUDIT-C World Health Organization (WHO) is comprised of the first three AUDIT questions which consider the consumption of alcohol, versus alcohol-related problems:How often do you have a drink containing alcohol? This is scored (0) Never; (1) Monthly or less; (2) 2 to 4 times a month; (3) 2 to 3 times a week; and (4) 4 or more times a weekHow many standard drinks containing alcohol do you drink in a typical day? This is scored (0) 1 or 2; (1) 3 or 4; (2) 5 or 6; (3) 7 to 9; and (4) More than 10How often do you have six or more drinks on one occasion? This is scored (0) Never; (1) Less than monthly; (2) Monthly; (3) Weekly; and (4) Daily or Almost Daily

This standardized 3-question approach offers useful standards for recording the frequency and levels of use. It is increasingly being used in studies involving alcohol use by pregnant women [34,36,37,38,39]. It has been used (and the language simplified) in research about FASD prevention in Indigenous (Australian Aborigine) communities in Australia [40]. It does not use the sex-specific number of drinks used in Canada’s low risk drinking guidelines for binge drinking; however, the Centers for Disease Control and Prevention (Atlanta, Georgia) have made adaptations in the standard drink size and cut-off points for use in the United States of America as the AUDIT 1–3 [41]. The AUDIT-C also does not distinguish among use before pregnancy, or before pregnancy recognition and current use, but the questions could be used, referencing these and other specific times in the perinatal period.

The Alcohol, Smoking and Substance Involvement Screening Test (ASSIST) (WHO) is a screening tool which has questions about alcohol and other substances. The ASSIST questionnaire asks about any use of alcohol and eight other substances in one’s life, and in the past three months, as well as a range of questions to determine problems with use (frequency of: Having a strong desire to use, use that led to health/social/legal/financial problems, failed to do what was expected because of use, tried and failed to cut down, etc.). The response card allows the patient to say never (not in the past 3 months), 1 to 2 times in the past 3 months, 1–3 times in one month, 1–4 times per week, daily or almost daily.

There are alcohol screening tools that have been validated with pregnant women such as the T-ACE [42], TWEAK [43,44], and 4Ps [45] which identify indicators of problems associated with alcohol use, and use by friends and family that may influence a women’s alcohol consumption. Some researchers question the usefulness of such approaches [35]. While they may be useful in the clinical context to know about problematic use in order to identify women who could benefit from a referral to treatment, for purposes of FASD prevention, it is important to know the basics of quantity, the frequency and level of use to better understand what constitutes harmful levels of exposure, and to inform messaging in preventive efforts with women.

#### 3.1.2. Questions Used for Surveillance of Alcohol Use by Pregnant Women in Canada

There has not been a consistent approach across Canadian provinces and territories to the collection of data on alcohol consumption and recording it on the antenatal record forms. The surveillance experts who participated in the April 2018 consultation reported three approaches undertaken by provinces/territories to collecting and documenting alcohol use in pregnancy: (1) Framing questions about alcohol use using basic yes/no questions to determine whether women were drinking before pregnancy, and/or in their current pregnancy; (2) Capturing information numerically, i.e., ask about: The maximum number of drinks per day, the number of drinks before pregnancy and currently, and/or the number of drinks per day, per week, and the maximum number of drinks per drinking day; or (3) Using a full checklist that includes: If women have ever consumed alcohol, drinks per week before pregnancy, drinks per week during pregnancy, quit date, scores from standardized screening tools, re-assessment dates based on risk and/or drug screens and test results [33].

Participants in this meeting expressed concern about overcoming key persistent barriers to collecting and recording this important information, including: The lack of time on the part of health care providers; the stigma associated with alcohol use by women in pregnancy, which affects both providers and women; and the lack of awareness on the part of women as to what comprises a standard drink [33].

### 3.2. Summary of Information Asked about Alcohol Use in the Pregnancy Cohorts

Questions about alcohol use have been asked differently, and with different levels of precision in each of the cohort studies. The cohorts used between two and 32 specific questions related to alcohol use. Table 1 summarizes the type of information gathered prior to, during and after pregnancy for each of the 12 cohorts. Eight cohorts included at least some information about alcohol use at all three general time periods.

As recruitment occurred after pregnancy recognition, all information about preconception alcohol use was collected retrospectively. The information about alcohol use was collected between one and six times over the longitudinal data collection. Two of the cohorts only asked about alcohol use at one time period. GESTE asked about alcohol use during pregnancy only retrospectively, at birth.

Table 2 includes the questions used to collect data about alcohol use during pregnancy, and the corresponding response categories. This table combines all the questions asked across the three trimesters, and only includes the seven cohorts for which specific questions asked were included in the catalog.

#### 3.2.1. Occurrence of Alcohol Use

Of the 12 cohort studies reviewed, eight include a variable about the occurrence of any alcohol use. For example, “In the last year, have you consumed any alcoholic drinks?” or “During the past week, did you drink alcohol?”. It is recommended that a question about the general occurrence of alcohol use be asked in a consistent manner, for the 3 months prior to becoming pregnant, as well as during each trimester of pregnancy, and during all post-partum follow up.

#### 3.2.2. Frequency of Alcohol Use

Ten of the 12 cohort studies collected data about the frequency of alcohol use during pregnancy, nine postpartum and six preconception. Only one cohort included an additional question about the frequency of alcohol use while pregnant, but prior to pregnancy recognition. Three cohorts used open text fields to gather the number of days a woman consumed alcohol within a specified time period. For example, 3D used the open-ended question, “How often did you drink alcoholic beverages?” and collected information for the number of drinks per day, week and month. AOB/F used the response category of “less than 1”, but then included two through seven as a continuous variable for questions about weekly drinking frequency. The remaining six cohorts used other categorical responses to collect frequency information. The number of response categories varied across the eight cohorts. Five of the cohorts included questions without a lowest response category of “none”. For example, two cohorts used the lowest category “Never, or less than 1 drink a month” and another “less than once a month” for questions about monthly drinking. Similar to overall occurrence, it is recommended that any frequency of alcohol use be asked about, in a consistent manner, for the 3 months prior to becoming pregnant, as well as during each trimester of pregnancy, and during any longitudinal follow up. Standardizing the reporting period (e.g., during the past three months), will allow for more accurate comparison across cohort data. Additionally, including a lowest response category of “none” will allow for analysis to separately investigate the effects of no and low-level alcohol use.

#### 3.2.3. Quantity of Alcohol Use

Ten of the 12 cohort studies collected data about the quantity of alcohol used during pregnancy, with eight cohorts collecting information about both frequency and amount. Three cohorts collected information about the quantity of alcohol use during preconception, and three postpartum (AOB/F and APrON collected both). The method for collecting the amount of alcohol varied across the cohorts, and most used an open response field to indicate a number of drinks within a specific time period.

The timeframe for the amount of alcohol used varied across the cohorts. 3D collected information about the number of drinks on each day using a past week recall for seven days. ABC asked about the total number of drinks over the entire pregnancy at the time of birth. Five cohorts asked questions about the number of drinks per occasion using questions similar to “On the days that you drank, how many drinks did you usually have?”, while three asked about amount of drinks per week. Similar to frequency, ABOB/F used response categories, but they were “Less than 1”, continuous two though four and “5 or more”. Similar to frequency, not separating no drinking from low-level drinking in the data collection limits the potential to investigate the differing effects of these two patterns of use.

In addition to the quantity of alcohol consumed overall, three cohorts asked questions about the different types of alcohol consumed. MIROS-C included specific information on beer, spirits and wine; MIREC on beer, liquor, white wine and red wine; and OBS asked about the number of times per week, and total number of drinks women consumed for beer, white wine, red wine, spirits/ liquor and other alcohol. MIROS-C also asked about the amount of alcohol in each drink, e.g., “If yes (you had any alcoholic drinks in the last 3 months), quantity (mL) per drink on average? Spirits.”

It is recommended that women be asked how many standard drinks they drink on a typical day when they are drinking, and that visual aids about what comprises a standard drink is shared, to ensure that the amount within a standard drink is understood. A lowest response category of “0” should be used, and should not be included as “less than one”. Again, this information should be collected using consistent questions for the 3 months prior to becoming pregnant, as well as during each trimester of pregnancy, and during any longitudinal follow up.

#### 3.2.4. Heavy Alcohol Use

Seven cohorts had questions specifically related to heavy alcohol use, and another four had responses for either the quantity or frequency questions that would provide this information. Of these, five used a cut off of five or more drinks. For example, FAMILY asked, “During this pregnancy, how many times have you consumed at least 5 or more drinks of alcohol in a day?” OBS and MIROS C used the cut off of four or more drinks per occasion, in line with the current Canadian low risk drinking guidelines for risky heavy alcohol use for women [6]. Two of the cohorts had “yes” or “no” response categories for heavy alcohol use, while the other five included some information about the frequency of their heavy alcohol use. It is recommended that women be asked: How often did they drink more than 3 drinks on one occasion? Using the response categories (0) Never (1) Less than monthly (2) Monthly (3) Weekly (4) Daily or Almost Daily, consistently in the identified before, during and after time periods.

#### 3.2.5. Quit Date

Three cohorts included information about when women quit drinking. Two of them included dates to capture information about when women who drank prior to pregnancy stopped using alcohol, and the other asked, based on a specified time period, “was your alcohol consumption about the same, or less, compared to your usual alcohol consumption?” As women may continue to drink at pre-pregnancy rates prior to pregnancy recognition, it is important to gather information about alcohol use during early pregnancy, and to identify a quit date, if relevant. It is recommended that women are asked in the first data collection period if their drinking prior to finding out they were pregnant was more, less or the same as after they found out. If drinking is identified as more or less, separate questions, as advised in previous sections, should be used to follow up about the quantity and frequency of alcohol use in the two time periods.

## 4. Discussion

There are challenges to gathering information about women’s alcohol use in the clinical context, related to barriers to reporting by women and to asking by health care providers. There are also barriers in establishing common Canadian standards to collect information across cohort studies. It is however important to know about alcohol use in pregnancy, in order to develop FASD prevention strategies based on accurate information about the prevalence, patterns and influences on women’s alcohol consumption. Accurate information about alcohol use in pregnancy also informs FASD diagnostic processes.

In this paper we have briefly outlined approaches to asking questions about maternal alcohol use employed by: (1) Evidence-based standardized alcohol screening tools, (2) the Canadian provinces and territories for the documentation of alcohol consumption for surveillance purposes, and (3) Canadian pregnancy cohort studies. Based on these three sources, we offer analysis and recommendations for a harmonization of data collection in the research context, which could contribute significantly to a more adequate understanding of alcohol use by women in the perinatal period to inform FASD prevention strategies.

Key elements about effective approaches asking about alcohol use that could be incorporated in pregnancy cohort research contexts, where confidentiality is ensured, are:Asking about frequency of use using standardized categories, such as those in the AUDIT-C ((0) Never (1) Monthly or less (2) 2 to 4 times a month (3) 2 to 3 times a week (4) 4 or more times a week). This allows for a comparison across cohorts, and with other studies that employ the commonly used AUDIT-C, and corrects the problem that some cohorts had the lowest response category of “one or less” or “less than once a month”; i.e., it did not allow for the separation of no drinking, and low-level drinking.Asking about the quantity of alcohol consumption using standardized categories such as those in the AUDIT-C which are (0) 1 or 2 (1) 3 or 4 (2) 5 or 6 (3) 7 to 9 (4) more than 10, or a continuous number that can be re-categorized as necessary. The use of visual tools that picture what comprises a standard drink has been found to be helpful when asking about quantity, as many women do not know about standard drink sizes [46]. These tools show pictures of standard amounts of various types of alcohol, which can help women see that any type of alcohol may affect the fetus, given that misconceptions prevail about the benefits of some types of alcohol (e.g., red wine). Using both low and high examples of the number of drinks, such as is in the AUDIT categories, makes room for those who do drink at risky levels to see that researchers are interested in, and can handle, hearing that more than 10 is a possibility.Asking about the number of drinks consumed on an occasion, as this has a bearing on the risk of adverse effects on the fetus, as well as on women’s health. It is recommended that “more than 3 drinks on an occasion” be used (or “4 or more”) to align with what is considered risky for all women in Canada’s Low Risk Alcohol Drinking Guidelines (LRDGs). In this way women will be able to see the consistency of any questions about alcohol that are being asked when they are accessing health care services generally, to those asked about consumption before, during and after pregnancy in the context of the pregnancy cohort studies. Learning about these LRDGs and reporting on alcohol consumption, referring to LRDGs guidance, is being recommended in alcohol screening with all men and women.Asking about alcohol consumption before pregnancy (using the same standardized response categories for frequency and quantity described above), changes in use since pregnancy recognition, and quit date are also important information that could be gathered uniformly in the pregnancy cohorts. This information is important to understand for several reasons. When linked to birth outcomes, it can provide critical information to understanding risk, for example, the risk associated with any heavy preconception use.It is also recommended that, no matter in what context that questions are asked about alcohol use, that this be done in a trauma informed and culturally sensitive way, and in a way that makes it safe for women to report harm reduction efforts, and not only abstinence. Women who have experienced trauma are more likely to drink during pregnancy [47], and experience more mental health issues [48]. Trauma-informed approaches are based in safety and collaboration [49], and culturally safe, strengths-based and wellness -oriented approaches are foundational to any discussions of alcohol with Indigenous girls and women [50,51]. Clinical guidelines on the topic of pregnancy and substance use are now making specific recommendations about using trauma informed, equity informed and culturally safe approaches, and such approaches are relevant for the research context as well [52].

Harmonizing and integrating data from multiple studies across Canada would provide a unique opportunity to document information regarding women’s drinking during pregnancy. The co-analysis of harmonized data would also facilitate any investigation of the factors and characteristics that are associated with women drinking during pregnancy, such as age, education, household income, ethnicity, drug use, and smoking (We are currently undergoing this analysis, and the results will be described in a separate paper).

The creation of a larger sample could also be used to investigate the outcomes and correlates of more rare behaviors, such as heavy binge drinking during pregnancy, and to identify women’s trajectories of alcohol use before, during and after pregnancy [30,31,32]. Harmonizing data sets that include birth outcomes and longitudinal data collection with infants and children will also allow for greater opportunities to measure the effects of different trajectories of alcohol use. However, the potential to harmonize cohort data is limited by the similarity of any information collected, and how it is collected.

## 5. Conclusions

We see the potential for a harmonization of the way inquiry about alcohol use in the perinatal period is done in pregnancy cohort research. A harmonized approach could allow for the generation of accurate information about alcohol use in pregnancy, that to date has not been well attained in clinical settings and through surveillance mechanisms. Such data on women’s alcohol use before, during and following pregnancy will be highly useful to inform FASD prevention strategies.

## Figures and Tables

**Table 1 ijerph-16-02019-t001:** Summary of information collected about alcohol use prior to, during and after pregnancy.

Cohort	Prior to Pregnancy				During Pregnancy				Postpartum			
	Occurrence	Frequency	Quantity	Heavy Drinking	Occurrence	Frequency	Quantity	Heavy Drinking	Occurrence	Frequency	Quantity	Heavy Drinking
3D ^1^	-	X	-	X	X	X	X	X	-	X	-	X
ABC ^2^	-	X	-	-	-	X	X	X *	-	X	-	-
AOB/F ^3^	X	X	X	X	-	X	X	X	X	X	X	X
APrON ^4^	X	X	X	X *	X	X	X	X *	X	X	X	X *
FAMILY ^5^	-	X	-	-	-	X	-	X	-	X	-	-
FPM ^6^	X	-	-	-	X	X	X	X	X	X	X	X
GESTE ^7^	-	-	-	-	X	-	X	X *	-	-	-	-
MIROS C ^8^	-	-	-	-	X	X	X	-	-	X	-	X
MIREC ^9^	-	-	-	-	X	X	X	X	-	-	-	-
Oak ^10^	-	-	-	-	X	X	X	X *	-	-	-	-
OBS ^11^	-	-	X	X	X	-	X	X	-	X	-	X
START ^12^	-	X	-	X	-	X	-	X	-	X	-	X

^1^ 3D Study—Design, Develop, Discover; ^2^ Aboriginal Birth Cohort; ^3^ All Our Babies and All Our Families; ^4^ Alberta Pregnancy Outcomes and Nutrition; ^5^ Family Atherosclerosis Monitoring in Early Life; ^6^ Feelings in Pregnancy and Motherhood; ^7^ Pregnancy and healthy child: A study of thyroid and environment; ^8^ Oxidative stress, fetal growth and programming of the metabolic syndrome and cardiovascular disorders; ^9^ Maternal-Infant Research on Environmental Chemicals; ^10^ Ottawa and Kingston Birth Cohort; ^11^ Ontario Birth Study; ^12^ South Asian Birth Cohort; * Although no question was specifically asked about heavy alcohol use, this information could be calculated based on daily quantity responses, or based on a categorical response to frequency (e.g., “more than 5 a day”).

**Table 2 ijerph-16-02019-t002:** Questions used to capture data about alcohol use and response categories, during pregnancy. *

	3D	ABC	AOB/F	FAMILY	FPM	MIROS C	OBS
Occurrence	N/A	N/A	Alcohol consumption after pregnancy recognition Yes No	N/A	During the past week, did you drink alcohol? Yes No	Have you had any alcoholic drinks in the last 3 months? No Yes Unknown	Have you consumed alcohol over the past 3 months? Yes No
Frequency	Since pregnant, number drink - days/week? [Open text] Since pregnant, number drink - total number days? [Open text]	Number of drinks (during pregnancy) [Open text]	Once you knew you were pregnant, how many days did you drink alcohol (on average)? Less than 1 2 3 4 5 6 7	During this pregnancy, how often do you drink alcohol in a month? Never, or less than 1 drink a month Once a month Between 2 and 3 times a month Once a week Between 2 and 3 times a week Between 4 and 6 times a week Everyday	During the past year/since last meeting, how often do you drink alcohol? Less than once a month Once a month 2 to 3 times/month Once a wee 2 to 3 times/week 4 to 6 times/week Everyday	If yes (you had any alcoholic drinks in the last 3 months), number of alcoholic drinks per week? Beer/Spirit/Wine [Open text]	Over the past 3 months, how often did you drink alcohol? 6 to 7 times a week 4 to 5 times a week 2 to 3 times a week Once a week 2 to 3 times a month About once a month Less than monthly Prefer not to answer
Quantity	On the days that you drank, since you have become pregnant, how many drinks did you usually have? [Open text]	During pregnancy, how often do you have a drink of beer, wine, liquor, or any other alcoholic beverage? Never, or less than 1 drink a month Once a month Between 2 and 3 times a month Once a week Between 2 and 3 times a week Between 4 and 6 times a week Everyday Greater than 5 drinks in a single day	How many drinks would you typically have when you drank (on average)? Less than 1 1 2 3 4 5 or more	N/A	In the last month, how often did you drink alcohol? Occasional drink 1 or 2 1–2 drinks a day 5+ drinks at one time Quit since pregnant Quit before pregnant Never drank Occasional & 5+ 1–2 drinks a day & 5+ Occasional drink, quit since pregnant 5+ and quit since pregnant Occasional and quit before pregnant 1–2 drinks/day, quit since pregnant 5+ and quit before pregnant Occasional; 5+, quit since pregnant Never drank	If yes (you had any alcoholic drinks in the last 3 months), quantity (mL) per drink on average? Beer/Spirit/Wine [Open text]	Currently, during your pregnancy, how many drinks did you have on average during a typical week? [Open text] Over the past 3 months, on average, how many drinks did you have during a typical week? [Open text]
Heavy use	Since pregnant, 5 or more drinks - days/week? [Open text] Since pregnant 5 or more drinks - days/month? [Open text] Since pregnant, 5 or more drinks - total number of days? [Open text]	[See quantity]	Since becoming pregnant (including before you knew you were pregnant), did you ever drink 5 or more drinks on any one occasion? Yes No	During this pregnancy, have you consumed at least 5 or more drinks of alcohol in a day? Yes No During this pregnancy, how many time have you consumed at least 5 or more drinks of alcohol in a day? [Open text]	During the past year/ since last meeting, how often did you have more than 5 drinks at one time? Never Less than once a month Once a month 2 to 3 times/month Once a week More than once a week	N/A	Currently, during your pregnancy, how often did you have four or more drinks at the same sitting or occasion? 1 or more times per month Less than once per month None Don’t know Prefer not to answer

* includes only the seven cohorts in the Research Advancement through Cohort Cataloguing and Harmonization (ReACH) Catalog that included the actual questions used in the research.

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
