# Peer review of "The Potential for Fetal Alcohol Spectrum Disorder Prevention of a Harmonized Approach to Data Collection about Alcohol Use in Pregnancy Cohort Studies"

_ijerph, 2019, doi:10.3390/ijerph16112019_

Round 1

Reviewer 1 Report

Thank you for the opportunity to review this study.  Asking women about how much alcohol they consume in pregnancy is important for the mother and child.  Whilst there is some interesting information in the paper it requires more focus and the specific aims more clarity.  If the aim is to identify the most useful way of measuring and comparing ways of asking about alcohol use in pregnancy, there should be some method of comparison between questions/ methods.  As it stands the paper is more an audit of methods used in some cohort studies and screening tools than a research question.

I have reviewed the article and outlined my suggestions below.

Introduction

The authors examined several cohort studies to determine the range of questions asked about alcohol use in pregnancy. They note that use of that the confidential research context (that is use of previously collected cohort studies) will be helpful, perhaps because of the anonymity (although this is not specifically stated), in that women are more likely to disclose their drinking patterns during pregnancy.   Please justify.

The authors also note that several research cohorts can be bought together to increase sample size so there is adequate representation in regional areas. However, how well these studies can achieve this is not discussed further.  That is, the underlying rationale for using these methods is not addressed by analysis in the paper in the paper.

I am unclear as to the purpose of the information about screening. The authors note (page 2 line 71) that this will define how questions about how alcohol use in pregnancy should be asked.  In the methods section, however, there is an overview of some of the screening tools that have been used, but no discussion about what might be considered ‘gold standard’.

Materials and methods

First para. What information is taken from the screening tools?  How is best practice identified?

What information is presented about provincial perinatal surveillance efforts? The stated aim “We use these findings to make recommendations on guidelines for data collection about women’s alcohol use by future pregnancy cohorts” is unclear. What cohorts are the authors referring to? How will the recommendations be made? Will the validity and reliability of measures be compared?

Page 3 Line 113 This para is unclear.  Identification of current practices would require an audit of particular time/place.

Results

The section on current measures of alcohol use in pregnancy in the cohort studies assessed is a description of the wording used in the questions and whether some measures could be combined to increase sample size on particular items. As noted a more critical analysis of this information is needed. Comments on the section on screening and surveillance are similar, each section should be a critical analysis of current practice and specific recommendations.

Author Response

Introduction

The authors examined several   cohort studies to determine the range of questions asked about alcohol use in   pregnancy. They note that use of that the confidential research context (that   is use of previously collected cohort studies) will be helpful, perhaps   because of the anonymity (although this is not specifically stated), in that   women are more likely to disclose their drinking patterns during   pregnancy.  Please justify.

Added a sentence to describing   confidentiality of the research context and its impact on safety to report.

The authors also note that several   research cohorts can be bought together to increase sample size so there is   adequate representation in regional areas. However, how well these studies   can achieve this is not discussed further.  That is, the underlying   rationale for using these methods is not addressed by analysis in the paper   in the paper.

A section on the underlying   rationale for using these methods has been added to the introduction.

I am unclear as to the   purpose of the information about screening. The authors note (page 2 line 71)   that this will define how questions about how alcohol use in pregnancy should   be asked. In the methods section, however, there is an overview of some   of the screening tools that have been used, but no discussion about what   might be considered ‘gold standard’.

There is no gold   standard for screening for basic alcohol use in pregnancy. A sentence is   added to this effect

Note the validated   tools for assessing problematic use in pregnancy were validated in 1994, and   the Provincial and Territorial (P/T) surveillance experts note that they not   well received by women and health care practitioners. Since we are not focussing   on problematic alcohol consumption, we made brief reference to this in the   paper.

Materials and methods

First para. What   information is taken from the screening tools?  How is best practice   identified?

We have described   the screening tools and questions only in this paper – an analysis of the   data elicited will be described in a later paper. We make recommendations   based on triangulating current practice and standardized screeners for the   general population.

What information is   presented about provincial perinatal surveillance efforts? The stated aim “We   use these findings to make recommendations on guidelines for data collection   about women’s alcohol use by future pregnancy cohorts” is unclear. What   cohorts are the authors referring to? How will the recommendations be made?   Will the validity and reliability of measures be compared?

The information   about provincial perinatal surveillance efforts is described in section 3.1.2.

We are referring to   the recommendations made in the discussion section, which are indented for   both the further questioning of women in the longitudinal studies now in   place, as well as any future cohorts funded. Specific recommendations have   also been added to the results section.

Page 3 Line 113 This   para is unclear.  Identification of current practices would require an   audit of particular time/place.

Wording clarified to   say we ‘present’ these two types of data collection and discuss them.

 Results

The section on current   measures of alcohol use in pregnancy in the cohort studies assessed is a   description of the wording used in the questions and whether some measures   could be combined to increase sample size on particular items. As noted a   more critical analysis of this information is needed. Comments on the section   on screening and surveillance are similar, each section should be a critical   analysis of current practice and specific recommendations.

We   reorganized the sections 3.1 and 3.2 to present the information on the   standardized screeners and the current perinatal surveillance efforts first   in an effort to more effectively make our argument of how these might inform   the data collection in the research context. Specific recommendations related   to the research context have now been added to each subsection of 3.2 as well   as being summarized in the discussion section

Reviewer 2 Report

This is a very well written paper, with a valid rationale and clearly presented methods and results (if somewhat atypical as they are not the standard results of statistical analyses, but more a review of data available and their format). The authors have extensively reviewed how prospective Canadian cohorts collected data on prenatal alcohol exposure, alongside what standardised tools are available and also how Canadian surveillance systems collect this information. 

Some minor comments follow.

- The authors push for using cohort data collected in a research context to inform prevention strategies for FASD, citing stigma and other barriers to collecting accurate self-reported data in clinical or surveillance contexts. What evidence do they have that the (potentially heavily selected) population samples participating in cohort studies will provide more valid estimates of prenatal alcohol consumption, and furthermore that these will also be generalisable? Have any efforts been made to triangulate or link the information from cohorts with the information from routine surveillance systems? What about the inclusion or exclusion of certain marginalised groups in the population and in particular ethnic minorities including First Nations? It is also not clear how data from a population based cohort study could directly help diagnose children with FASD or support them (as claimed at the end of the first paragraph of the Discussion) - often there are ethical issues around identifying or diagnosing participants in a cohort study, and what about the informed consent? 

- The authors should include mention of the biometric perfomance of different questionnaires and standardised tools for assessing prenatal alcohol exposure, for a complete picture on the 'strengths and limitations' of each tool as claimed in the second paragraph of the Discussion - what is the gold standard? What is known about how each questionnaire/tool fares compared to the gold standard? What is the reliability of each questionnaire/tool? This is an important aspect to consider when deciding which tool to adopt going forward, but also in choosing how to harmonize current data from cohort studies where standard validated tools like the AUDIT-C aren't available.

- Since cohort studies use prenatal alcohol exposure data to provide estimates of risk for various offspring outcomes (not just to report on the prevalence of drinking behaviour), the authors should mention how some tools/questionnaires could bias these estimates, based on their known limitations (eg a tool what is known to lead to systematic underreporting/overreporting of alcohol use could bias estimates in this or that direction etc....).

- A more extensive discussion of the issues of cultural sensitivity, diversity and inclusion in relation to prenatal alcohol exposure questions should be included.

- The authors don't comment on collecting information about partner's drinking behaviour before and during pregnancy. This is another important and related set of variables, and could per se influence offspring risk, but at present few datasets include these data (to this effect, see https://www.ncbi.nlm.nih.gov/pubmed/30055422). 

Author Response

Reviewer 2:

- The authors push for   using cohort data collected in a research context to inform prevention   strategies for FASD, citing stigma and other barriers to collecting accurate   self-reported data in clinical or surveillance contexts. What evidence do   they have that the (potentially heavily selected) population samples   participating in cohort studies will provide more valid estimates of prenatal   alcohol consumption, and furthermore that these will also be generalisable?   Have any efforts been made to triangulate or link the information from   cohorts with the information from routine surveillance systems?

What about the   inclusion or exclusion of certain marginalised groups in the population and   in particular ethnic minorities including First Nations? It is also not clear   how data from a population-based cohort study could directly help diagnose   children with FASD or support them (as claimed at the end of the first   paragraph of the Discussion) - often there are ethical issues around   identifying or diagnosing participants in a cohort study, and what about the   informed consent? 

While the data will   not likely provide a more valid estimate of prenatal alcohol consumption,   they have the advantage to provide the opportunity to collect longitudinal   information on alcohol consumption: before, during and after pregnancy. While   the information collected will not be fully generalizable, cohorts are more   useful to investigate associations. This has been addressed in the   introduction.

To our knowledge no   efforts have been made to triangulate the data from clinical surveillance and   research sources.  Unfortunately, we do   not have good data from either of these sources, and that is why we saw it a   beneficial to prepare an article that offers recommendations for improvement   in the research context. (We are also working with the P/T groups in a   national committee)

We are preparing a   second article on the findings from the research cohorts, by demographic   status and agree this is very important.

We changed the   sentence to show that we are not using this data in diagnostic efforts, but   simply that knowing of maternal use is a criterion of FASD diagnosis and that   ultimately knowing of levels, timing and frequency of alcohol consumption   will be important in the long run.

- The authors should   include mention of the biometric performance of different questionnaires and   standardised tools for assessing prenatal alcohol exposure, for a complete   picture on the 'strengths and limitations' of each tool as claimed in the   second paragraph of the Discussion - what is the gold standard? What is known   about how each questionnaire/tool fares compared to the gold standard? What   is the reliability of each questionnaire/tool? This is an important aspect to   consider when deciding which tool to adopt going forward, but also in   choosing how to harmonize current data from cohort studies where standard   validated tools like the AUDIT-C aren't available.

We cannot provide   the biometric performance, as the screeners such as the AUDIT C have not yet   been validated with pregnant women, only the tools assessing problem use such   as the TWEAK have. The TWEAK was validated in the 1990s when, for example, it   was more acceptable to ask women if others have criticized them for   their use, rather than asking them directly of their own assessment of how   much their alcohol use had interfered in their lives. We see that pregnant   women are able to report on their alcohol use with the same basic screening   questions, as all women are, and that there are benefits to keeping the   questions consistent, especially between the preconception and pregnancy   periods.  We are hoping that the AUDIT   C, with efforts in Canada and Australia will become the gold standard.

- Since cohort studies   use prenatal alcohol exposure data to provide estimates of risk for various   offspring outcomes (not just to report on the prevalence of drinking   behaviour), the authors should mention how some tools/questionnaires could   bias these estimates, based on their known limitations (eg a tool what is   known to lead to systematic underreporting/overreporting of alcohol use could   bias estimates in this or that direction etc....).

We can only advocate   for best practice in asking about alcohol use in the research context, where   over and under reporting is less likely – and hope that this will improve   understanding of how/if alcohol use is associated with various outcomes

The data from FASD   diagnostic clinics which is now being consolidated in a national database,   will hopefully serve to illuminate the linkage between levels and patterns of   alcohol use and specific alcohol-related impairments.

- A more extensive   discussion of the issues of cultural sensitivity, diversity and inclusion in   relation to prenatal alcohol exposure questions should be included.

We added a point in   the discussion in this and agree it needs always to be emphasized

- The authors don't   comment on collecting information about partner's drinking behaviour before   and during pregnancy. This is another important and related set of variables,   and could per se influence offspring risk, but at present few datasets   include these data (to this effect, see   https://www.ncbi.nlm.nih.gov/pubmed/30055422). 

We agree that is   important information to collect, and we are currently preparing research   summaries for the CanFASD Research Network on this topic. We felt it beyond   the scope of this paper 

Reviewer 3 Report

Well done to the authors. This is a well written and conceptualised study that has enormous potential to inform both policy and research. However there needs to be more said about this. For example, is standardisation possible in the clincal practice context? If so, how might this be addressed at a national level? What can be learned from efforts elewhere around the world e.g. the Australian context? Measuring risk in a standardised way can inform prevention efforts how? This should be addressed explicitly in the introduction and discussion sections.

Some further points for the authors to address:

Section 2.2 Please specify how both the structured screening tools and questions used in Canadian surveillance methods were identified. E.g. was a key word search conducted in the former and stakeholder consultations in the later?

The results section 3.1.1 and beyond (inclusive of Table 2) is not clear for the reader. In each section a number of cohorts are identified (e.g. 8 in 3.1.1, 10 in 3.1.2 and so on) however only 7 cohorts exist in Table 2. This needs clarity.

Lines 245 through 253: It is clear how the information is collected, but not how it is documented. Please address.

Line 305 – Once again, more needs to be said about how this might happen as per the initial comments.

I support accepting this paper for publication once the above comments are addressed.

Author Response

Reviewer 3:

Introduction

Is standardisation   possible in the clinical practice context? If so, how might this be addressed   at a national level? What can be learned from efforts elsewhere around the   world e.g. the Australian context? Measuring risk in a standardised way can   inform prevention efforts how? This should be addressed explicitly in the introduction   and discussion sections.

Standardization in   the clinical context is theoretically possible, but not realistic it requires   all provinces and territories and other governments to agree to a common   approach. We are working with the P/T surveillance experts on evidence   informed, standardized questions related to cannabis use in pregnancy and   hope there will be opportunity to discuss the importance of evidence   informed, standardized approach to the alcohol questions as well.  Of course, even if we have standardized   questions there remain the challenges of getting them asked and answered in   the clinical context, which to date has been an elusive goal.

Ways in which the   harmonization can improve the evidence to inform prevention efforts has been   added to the introduction.

Materials and methods

Section 2.2 Please   specify how both the structured screening tools and questions used in   Canadian surveillance methods were identified. E.g. was a key word search   conducted in the former and stakeholder consultations in the later?

Additional   information about the methods used to identify screening tools and Canadian   surveillance methods has been added to the Materials and Methods section.

 Results

The section on current   measures of alcohol use in pregnancy in the cohort studies assessed is a   description of the wording used in the questions and whether some measures   could be combined to increase sample size on particular items. As noted a   more critical analysis of this information is needed. Comments on the section   on screening and surveillance are similar, each section should be a critical   analysis of current practice and specific recommendations.

We   reorganized the sections 3.1 and 3.2 to present the information on the   standardized screeners and the current perinatal surveillance efforts first   in an effort to more effectively make our argument of how these might inform   the data collection in the research context.    Specific recommendations related to the research context have now been   added to each subsection of 3.2 as well as being summarized in the discussion   section

The results section   3.1.1 and beyond (inclusive of Table 2) is not clear for the reader. In each   section a number of cohorts are identified (e.g. 8 in 3.1.1, 10 in 3.1.2 and   so on) however only 7 cohorts exist in Table 2. This needs clarity.

This is explained in   section 2.2, and after the in-text introduction of the table in 3.2.1. Only   seven cohorts included information about the actual question used to gather   the data in the ReACH Catalogue, the remaining five only include data labels   indicating the content of the variable, not the actual question. A note has   been added to the table for clarification.

Lines 245 through 253:   It is clear how the information is collected, but not how it is documented.   Please address.

We have noted this in the   Methods section and in the Results section 3.1.2. The P/T get physicians to   document information on the antenatal records, and the P/T representatives reported   to the Centre of Excellence as to what info they gather and the challenges in   getting this data.

Line 305 – Once again,   more needs to be said about how this might happen as per the initial   comments.

Additional information   on the potential benefits of data harmonization and co-analysis have been   added to this section.